# Alleviation of Catching and Crating Stress by Dietary Supplementation of *Bacillus subtilis* in Pekin Ducks

**DOI:** 10.3390/ani12243479

**Published:** 2022-12-09

**Authors:** Helen Mitin, Idrus Zulkifli, Muhammad Hazim Che Jamri, Nur Athirah Zamzuri, Nurain Aliah Samian, Aimi Nabilah Hussein, Yong Meng Goh, Awis Qurni Sazili

**Affiliations:** 1Institute of Tropical Agriculture and Food Security, Universiti Putra Malaysia (UPM), Serdang 43400, Malaysia; 2Department of Veterinary Services, Federal Government Administrative Centre, Putrajaya 62630, Malaysia; 3Department of Animal Science, Faculty of Agriculture, Universiti Putra Malaysia (UPM), Serdang 43400, Malaysia; 4Department of Preclinical Sciences, Faculty of Veterinary Medicine, Universiti Putra Malaysia (UPM), Serdang 43400, Malaysia; 5Halal Products Research Institute, Universiti Putra Malaysia (UPM), Putra Infoport, Serdang 43400, Malaysia

**Keywords:** probiotic, stress, fear, behaviour, growth performance, catching and crating, Pekin ducks

## Abstract

**Simple Summary:**

Probiotics have been primarily used as feed additives to improve poultry growth performance and health. This study determined the effects of probiotic supplementation on the physiological stress response and fear-related behaviour in ducks subjected to catching and crating. Results showed that probiotics, as measured by serum levels of corticosterone, heat shock protein 70, creatine kinase, triglyceride, and heterophil to lymphocyte ratios, alleviated stress following crating. Fear-related behaviours in crated birds were dampened by probiotic supplementation.

**Abstract:**

Catching and crating may elicit stress and fear reactions in poultry because the procedures involve human contact and exposure to a novel environment. This study determined the effects of dietary probiotic supplementation on physiological stress, underlying fear, and growth performance of Pekin ducks subjected to catching and 4 h of crating. The study used a 2 × 2 factorial arrangement; the main factors were diet (basal or basal + probiotic) and crating durations (0 or 4 h). From 1 to 21 days of age (doa), birds were fed a basal or basal + probiotic (CLOSTAT^®^ (*Bacillus subtilis*) (Kemin Industries, Inc., Des Moines, IA, USA), 1 g/kg) diet. At 21 doa, an equal number of ducklings from each dietary group were caught and crated for 4 h or left undisturbed in the home pens. Birds were examined for serum corticosterone (CORT), heat shock protein (HSP) 70, creatine kinase (CK), triglyceride (TG), glucose (GLU), cholesterol (CHOL), and lactate (LAC) concentrations, heterophil to lymphocyte ratios (HLR), tonic immobility (TI) duration, open-field (OF) test, body weight (BW), and feed conversion ratios (FCR). Diet had no significant (*p* > 0.05) effect on CORT among the non-crated ducks. However, after catching and crating, birds fed the control diet had significantly (*p* < 0.05) higher CORT than their probiotic-supplemented counterparts. Catching and crating significantly (*p* < 0.05) elevated HSP70, HLR, GLU, and CHOL but reduced TG in ducks. Birds fed the probiotic-supplemented diet showed significantly (*p* < 0.05) lower HSP70, HLR, TG, and CK than those fed the control diet. Probiotic-supplemented ducks showed reduced fear-related behaviours, including TI durations, ambulation latency, and body shaking. Diet had a negligible effect on body weights and FCR of ducks at 21 doa. In brief, catching and crating for 4 h augmented Pekin ducks’ physiological stress and fear reactions, and supplementing birds with probiotics was beneficial in ameliorating these detrimental effects.

## 1. Introduction

There is a growing interest in gut microbiota due to its involvement in various human and animal bodily functions. It has been suggested that the gut microbiota profoundly impacts host phenotypes, presumably through the immune system, the nervous system, and endocrine pathways [1]. Gut microbiota may affect emotional behaviour, memory capacities, cognitive abilities, and behaviour in humans and animals [2,3,4]. It is increasingly recognised that gut microbiota may have significant implications for farm animal well-being [5,6]. Supplementing feed or drinking water with probiotics is one of the most common strategies to modify the gut microbiota composition of poultry. Probiotics can be defined as “live strains of strictly selected microorganisms which, when administered in adequate amounts, confer a health benefit on the host” [7]. They maintain a healthy bacterial balance in the gastrointestinal tract through competitive exclusion, a method whereby beneficial bacteria compete for attachment sites and nutrition in the intestines to exclude potentially dangerous bacteria [8].

In broilers, most probiotic investigations have focused on growth performance, immunity, gut health, carcass traits, and meat quality [9,10,11]. There is considerable evidence that probiotics are beneficial in alleviating heat stress in poultry [12,13,14]. However, the effects of probiotics on response to non-thermal stressors in poultry are limited. Probiotic supplementation has been shown to reduce distress calls in turkeys [15] and emotional reactivity in quails [16]. Zakari et al. [17] reported that probiotics attenuated underlying fearfulness and behavioural vigilance in broiler chickens. The effect of probiotics on physiological stress and fear responses in ducks is unknown. In the present study, we investigated the effects of probiotic (*Bacillus subtilis* at 1 g/kg diet) supplementation on physiological stress and underlying fearfulness in Pekin ducks subjected to catching and crating. Catching and crating, a routine practice in poultry production, involves human contact and exposure to novelty and thus may cause fear and stress to the birds [18,19,20,21,22]. Serum levels of corticosterone (CORT), heat shock protein (HSP) 70, and heterophil to lymphocyte ratios (HLR) were measured to evaluate physiological stress response. This study also examined how probiotic supplementation affects blood biochemistry (serum levels of creatine kinase (CK), total protein (TP), triglycerides (TG), glucose (GLU), cholesterol (CHOL) and lactate (LAC)) in crated ducks. We used tonic immobility (TI) [23,24] and open field (OF) [25] tests to estimate underlying fearfulness. The effect of probiotics on the growth performance of ducks during the starter phase was also determined.

## 2. Materials and Methods

All ducks were managed according to the Code of Practice for the Care and Use of Animals for Scientific Purposes by the Research Management Centre, Universiti Putra Malaysia. All experimental methods were approved by the Institutional Animal Care and Use Committee, Universiti Putra Malaysia (AUP–R048/2020).

### 2.1. Birds, Husbandry, and Housing

A total of 400-day-old Pekin male ducklings were obtained from a local duck breeder farm. On day 1, the ducklings were weighed and allocated randomly to 20 floor pens in groups of 20. Each of the concrete floor pens was covered with approximately 8 cm of wood shavings. The pens were in a conventional open-sided house with cyclic temperatures (minimum, 24 °C; maximum, 34 °C). Each pen contained two automatic bell drinkers and two tube feeders. The birds were provided starter (2950 kcal ME/kg; 21% crude protein) and grower (3050 kcal ME/kg; 19% crude protein) diets from days 1 to 14 and 15 to 21, respectively. Birds were given access to potable water and feed ad libitum. The lighting programme gradually decreased from 23 light: 1 dark to the first 3 days of age, then 16 light: 8 dark until 21 days.

### 2.2. Experimental Design

From days 1 to 21, an equal number of ducklings were fed either a basal diet (control) or basal diet + a commercial probiotic (CLOSTAT**^®^**, Kemin Industries Inc., Des Moines, IA, USA) at 1 g/kg diet. The probiotic contained *Bacillus subtilis* (2.2 × 10^8^ Colony Forming Units/g) as the dry direct-fed microbial. The freshly prepared feed was stored in a covered bin and kept in a cool and dry place. According to the manufacturer, the product is stable in feed for up to 3 weeks if stored as recommended. Each dietary group consisted of 10 replicate pens. On day 21 (08:00 am), five ducks from each pen were randomly selected, gently removed with minimum disturbance to flock mates, carried by their legs in an inverted manner and placed in plastic crates (0.80 m (length) × 0.60 m (width) × 0.31 m (height)) with 10 birds in each crate. The total number of birds sampled from each dietary group was 50. The crates were moved to another room (no visual contact with other birds) and left stationary for 4 h. The remaining ducks were left undisturbed in their home pens.

### 2.3. Blood Sampling

Immediately following crating, 15 birds from each dietary group were randomly selected for blood sampling. Fifteen birds from each dietary group that were not crated were randomly selected, and their blood samples were also collected. Approximately 3 mL of blood samples were collected by puncturing the medial metatarsal vein of the ducks. The ducks were caught and sampled one immediately after another. The catching and bleeding procedure did not exceed 1 min and should not influence circulating levels of corticosterone [26,27]. The blood samples were collected in plain (for serum) and ethylenediaminetetraacetic (EDTA)-coated (for whole blood) tubes. The blood samples for serum were centrifuged at 3000× *g* for 20 min at 4 °C, serum separated and stored at −80 °C awaiting analysis for CORT, HSP70, CK, TP, TG, GLU, CHOL, and LACT.

### 2.4. Measurement of Blood Parameters

According to manufacturer recommendations, CORT was determined using commercially available high-sensitivity EIA kits (Immunodiagnostic Systems Limited, Tyne & Wear, UK). Cross-reactivity of the corticosterone antiserum was less than 6.7% and 7.8%, respectively, and the detection limit was 27 ng/mL. The HSP70 determination was performed using a commercial ELISA kit (Cat. No. 201-16-0033, Shanghai Sunred Biological Technology, Shanghai, China) according to manufacturer recommendations. All samples were run in the same assay to prevent inter-assay variability. An automatic analyser (Hitachi 902, Tokyo, Japan) was used to determine serum levels of CK, TP, TG, GLU, CHOL, and LACT. All the reagents used were obtained from Roche (Roche Diagnostics, Basel, Switzerland). Blood smears were prepared using Wright’s stain, and H and L were counted to a total of 60 cells [28].

### 2.5. Tonic Immobility Test

Immediately following crating, 15 ducks (birds that were not blood sampled) from each dietary group were randomly selected for the tonic immobility (TI) test. Fifteen birds from each dietary group that were not crated were randomly selected and also subjected to TI test. Each individual was gently caught with both hands, held in an inverted manner, and carried to a separate room for TI measurements. A modification of the procedure described by Benoff and Siegel [29] was used. TI was be induced as soon as the bird arrived in the separate room by gently restraining it on its right side and wings for 15 s. The experimenter then retreated approximately 1 m and remained within sight of the bird but make no unnecessary noise or movement. Direct eye contact between the observer and the duck was avoided because it may prolong TI duration [24]. A stopwatch was started to record latencies until the bird righted itself. The restraining procedure was repeated if the bird righted itself in less than 10 s. If TI was not induced after three attempts, the duration of TI was considered 0 s. The maximum duration of TI allowed was 600 s.

### 2.6. Open-Field Test

Immediately following crating, 15 ducks (birds that were not blood sampled and tested for TI) from each dietary group were randomly selected for OF test. Fifteen birds from each dietary group that were not crated were randomly selected and also subjected to OF test. Birds were tested individually in an OF arena with dimensions of 2.0 m × 3.0 m. The arena was made of concrete flooring and was subdivided into four equal zones by two perpendicular fictive lines (Figure 1), according to Arnaud et al. [30]. The individual bird was always placed in Zone 1 of the arena, and their behaviours were video recorded for 5 min. The behaviour patterns were scored in focal sampling as follows: latency to the first immobilisation (LA), number of zones visited (ZV), duration of mobility (DM), number of pecks directed to the floor or the walls (P), vocalisations (V), and body shaking (BS).

### 2.7. Statistical Analysis

The structure of the experiment was based on a completely randomised design. Statistical analysis was carried out by using the General Linear Models (GLM) methodology of the Statistical Analysis System (SAS) 9.4 (2012). Individual ducks served as the experimental unit for all the parameters except body weight, feed intake, and FCR. The body weight, feed intake, and FCR data were analysed using *t*-test. Data were tested for normality, linearity, and homogeneity of variance. Transformations were explored when data did not meet the assumptions for parametric statistics. Diet, crating treatment, and their interactions were considered the main effects of the data analyses. When interactions between main effects were significant, comparisons were made within each factor. Where a significant treatment effect was found, post hoc comparisons of treatments were performed using Tukey’s HSD test. A *p*-value of less than or equal to 0.05 was considered significant. Data were expressed as the mean of each group ± SEM.

## 3. Results

### 3.1. Growth Performance

Diet had a negligible effect on weight gain, feed intake, and FCR from days 1 to 21 (Table 1).

### 3.2. Serum Corticosterone and Heat Shock Protein 70 Levels and Heterophil to Lymphocyte Ratios

Table 2 shows the results of CORT, HSP70, and HLR. The interaction between diet and crating treatment were only significant for CORT. Diet had no significant effect on CORT among the non-crated ducks (Table 3). However, birds fed the control diet showed significantly higher CORT than those provided a probiotic diet following catching and crating. The non-crated ducks had significantly lower HSP70 than their crated counterparts. Supplementing ducks with probiotics significantly reduced HSP70 compared to those fed the control diet. HLR were significantly affected by diet and crating treatment. A significantly lower HLR was noted in the probiotic-supplemented ducks compared to those fed the control diet. The crated ducks showed significantly higher HLR than the non-crated birds.

### 3.3. Blood Biochemistry

The effects of diet and crating treatment on serum biochemistry parameters are shown in Table 4. There was no significant interaction between diet and crating treatment for all the serum biochemistry parameters studied. The ducks fed the basal diet had significantly higher CK and TG than the probiotic-supplemented birds. Crating significantly lowered TG and GLU but elevated CHOL when compared to the non-crated ducks. Neither diet nor crating treatment had a significant effect on TP or LAC.

### 3.4. Tonic Immobility and Open-Field Test

The diet and crating treatment interaction for TI duration was not significant (Table 5). The probiotic-supplemented ducks showed significantly shorter TI duration than their control diet counterparts. Crating treatment had a negligible influence on TI duration. The effects of diet and crating treatment on OF test are presented in Table 6. There were significant interactions between diet and crating treatment for LA and BS. Following crating, the probiotic-supplemented ducks had shorter LA and lesser BS than birds fed the basal diet (Table 7). On the other hand, diet had no significant effect on LA and BS among the non-crated ducks. Supplementing probiotics to ducks increased ZV compared to those provided the control diet. Diet had no significant effect on DM, P, and V. Crated ducks showed longer DM, lesser ZV, and higher V than their non-crated counterparts. Both crated and non-crated birds had similar P.

## 4. Discussion

Probiotics have been demonstrated to be useful in maintaining or restoring healthy microbiota, limiting pathogen attachment to the intestinal wall, reducing inflammation, and maintaining the integrity of the intestinal barrier in poultry [31,32,33]. However, the effects of probiotics supplementation on the growth performance of ducks have been inconsistent. Lokapimasari et al. [34] reported that adding probiotics in the feed or drinking water enhanced the weight gain and feed efficiency of ducks. On the contrary, Kandir and Yardimci [35] reported otherwise. The present findings suggest that probiotics (*Bacillus subtilis* at 1 g/kg diet) supplementation had a negligible effect on ducks’ weight gain, feed intake, and FCR from days 1 to 21. Differences in the bacterial strains, concentrations of the probiotics, diet composition, and animal models used may be associated with conflicting results [6,36].

Crating ducks for 4 h in the present study exposed them to a number of potential stressors, such as food and water deprivation, novelty, and social tension. Hence, it is expected that the ducks had elevated CORT and HLR following 4 h of crating. Kannan and Mench [37] and Chloupek et al. [38] reported that plasma corticosterone concentrations of broiler chickens peaked at 3 and 4 h, respectively, following crating. However, work in chickens suggested that the elevated circulating levels of corticosterone following catching and crating declined with transport distance [22]. The effect of catching and crating on HLR has been demonstrated in broilers by Kannan and Mench [37] and Zulkifli et al. [20]. Elicitation of adrenocortical activity is known to precede heterophilia (or neutrophilia) and lymphopenia [39].

The effects of thermal and non-thermal stressors on HSP expression in chickens are well established [19,40,41,42]. These proteins, as chaperones, play a profound role in protecting animals against stress by re-establishing normal protein conformation and, thus, cellular homeostasis [43]. Heat shock protein expression has been used as an indicator of environmental stress in poultry [42,44,45,46]. The current results concur with those in broilers [19] that catching and crating induced HSP70 response. This is the first study demonstrating the influence of a non-thermal stressor on HSP70 expression in ducks.

The blood functions as the carrier of nutrients, metabolic wastes, and the pathway of humoral transmission [47]. Thus, the blood biochemical parameters would reflect the body’s physiological state. There has been limited research on blood biochemical responses to crating alone in poultry. Chloupek et al. [21] subjected chickens to 4, 8, and 12 h of crating and concluded that the duration of crating period was negatively correlated with the plasma levels of lactate dehydrogenase, cholesterol, triglycerides, glucose, and lactate. The present findings showed that 4 h of crating increased CHOL, reduced TG and GLU, and had no effect on TP and LAC. These discrepancies could be attributed to variations in species, and crating densities and durations. The noted increase in CHOL is in agreement with metabolic changes associated with stress in poultry [48]. Chloupek et al. [48] reported that the concentration of plasma triglycerides declined after 4 h of crating due to stress-induced lipolysis. According to Zhangh et al. [49], blood glucose levels of broiler chickens rose after 45 min of transport due to hepatic glycogen breakdown but declined dramatically after long-term transportation. During stress, catecholamines are produced, and glucose is released into the blood. However, a long crating period may cause glucose storage fatigue, thus reducing glucose concentration [50].

There is growing evidence that the gut microbiota interacts with the brain via the neurological, immunological, and endocrine systems to control brain activity and behaviour [51]. One of the earliest studies on the effect of gut microbiota on HPA axis during stress was by Sudo et al. [52]. The authors reported that the elevated plasma adrenocorticotropic hormone and corticosterone levels in mice were reversed by reconstitution with *Bifidobacterium infantis*. Thus, probiotic supplementation may modify the gut microbiota composition and alter stress response concomitantly. While there has been substantial work on the effect of probiotics on heat stress in poultry [53,54], there is a paucity of information on the influence of the supplement on response to non-thermal stressors. The present findings suggest that, as measured by CORT and HLR, probiotics supplementation can alleviate stress attributed to catching and crating in ducks. Chen et al. [55] reported that zebrafish fed *Rhizopus oryzae*-fermented soybean tempeh had improved gut microbiota composition and brain-derived neurotrophic factor (BNDF) expression during pre-stress and post-stress conditions. BDNF is a crucial neurotrophic factor in the brain and can penetrate the blood–brain barrier [56]. It can increase neuroplasticity, aid in nerve development and differentiation, and directly impact cognitive processes, including memory and adaptability in the brain [57]. The underlying mechanisms for the antistress effect of probiotics in poultry merit further investigations.

Heat shock proteins are expressed under normal conditions and elicited when the tissues are damaged by environmental perturbations such as heat, feed deprivation, overcrowding, social isolation, or transportation [41,42,58,59,60]. Hence, the increase in HSP70 in stressful conditions is used as a physiological measure of stress in poultry. The present findings suggest that supplementing ducks with probiotics can dampen the HSP70 response to catching and crating in ducks. Compared to controls, the lower HSP70 expressions in the probiotic-supplemented ducks suggest that catching and crating were not recognised as stressors at the cellular level. Cryan et al. [61] indicated that probiotics-elicited microbial changes could protect against stress-induced protein damage via the microbiota–gut–brain axis. Elevated creatine kinase activity has been associated with muscle damage due to disrupted muscle cell membrane function and permeability [62]. Al-Aqil and Zulkifli [44] and Zhang et al. [49] reported that road transportation elevated serum levels of CK in broilers. The noted reduced CK in birds provided probiotics suggests that the supplement may lower enzyme activity and act as a protective agent for the liver and muscles against damage factors. The present findings agree with Kalavathy et al. [63] and Ashayerizadeh et al. [64] that probiotic supplementation may reduce TG in poultry. The effect of probiotics on TG is associated with lower lipid absorption or higher lipid catabolism [65].

Fear and stress are not synonymous, but fear-related behaviour is closely associated with the stress response that is regulated by the hypothalamic–pituitary–adrenal (HPA) [44]. LeDoux [66] suggested that animals may experience fear when an emotional stressor stimulates the HPA axis. Although crating is stressful for chickens [21,37,51], the effect on fear response has been inconsistent. Zulkifli et al. [67] reported that crating for 10 min prolonged TI duration in chickens. On the contrary, Cashman et al. [68], Mills and Nicol [69], and Mielnik et al. [70] showed that holding broilers and pullets in crates for several hours had a negligible influence on TI duration. The present findings suggested that crating for 4 h did not influence TI duration in ducks. The conflicting findings could be associated with the method of catching, duration of crating, and environmental conditions.

The OF test involves removal from the home environment, abrupt isolation, and exposure to a new, primarily barren environment that is frequently larger and brighter illuminated than the home cage [71]. Fear of novelty is one element that influences OF behaviour, but there are many more, including social motivation, exploratory behaviour, and territorial marking [24]. Subjecting ducks to catching and crating, as measured by DM, ZV and V, heightened fearfulness. The noted significantly longer LA and DM, and higher V displayed by the crated ducks indicate that they were more fearful than their non-stressed counterparts. The higher degree of fear-indicating behaviours in the stressed ducks could be associated with social reinstatement. Marin et al. [72] and Guzman et al. [73] indicated that social reinstatement could be linked to fear and stress reactions. The lower ZV showed by the crated ducks can be interpreted as a reduced exploration of a novel environment. The relationship between fear and exploration in novel situations has been recently characterised by Meuser et al. [74].

We demonstrate for the first time that probiotic supplementation, as measured by TI and OF tests, may attenuate underlying fearfulness in ducks. In the present study, the probiotics-supplemented ducks had shorter TI duration than their control counterparts. Similarly, Zakari et al. [17] reported that broiler chickens fed 0.55 g/kg of probiotics (Bactofort^®^) showed shorter TI durations than those provided basal diets. Out of the six behavioural parameters recorded in the OF test, two measures responded to the probiotic supplementation. Interestingly, there were significant interactions between diet and crating treatment for LA and BS in the OF test. When subjected to catching and crating, the ducks supplemented with probiotics showed significantly shorter LA and more frequent BS than controls. On the contrary, both groups of birds had similar LA and BS under non-stressed conditions. Higher rates of BS can be interpreted as suggesting heightened fear in a novel environment [75]. Puetz et al. [76] showed that the gut microbiome is associated with the fear of humans in the red jungle fowl. The ability of gut microbiota to communicate with the brain, possibly through neural, endocrine, and immune pathways, may modulate the behaviour of human beings and animals [77,78]. Probiotics benefit the host by improving GIT activities, thus regulating microbiota–gut–brain activity, including emotional and behavioural reactions [5,16,79].

## 5. Conclusions

In conclusion, our findings confirm that catching and crating for 4 h can cause stress and fear in ducks. The current results suggest that probiotics, *B. subtilis* (CLOSTAT^®^), at the 1 g/kg level, are beneficial in dampening stress and fear responses in ducks. Compared to controls, probiotics-supplemented broilers show reduced serum levels of corticosterone, heterophil to lymphocyte ratios, and serum levels of heat shock protein 70. Probiotics also have a favourable effect on tonic immobility duration and behaviours recorded in the open-field test. Overall, the current findings indicate that supplementing probiotics could be a practical management strategy to improve the well-being of Pekin ducks during catching and crating.

## Figures and Tables

**Figure 1 animals-12-03479-f001:**
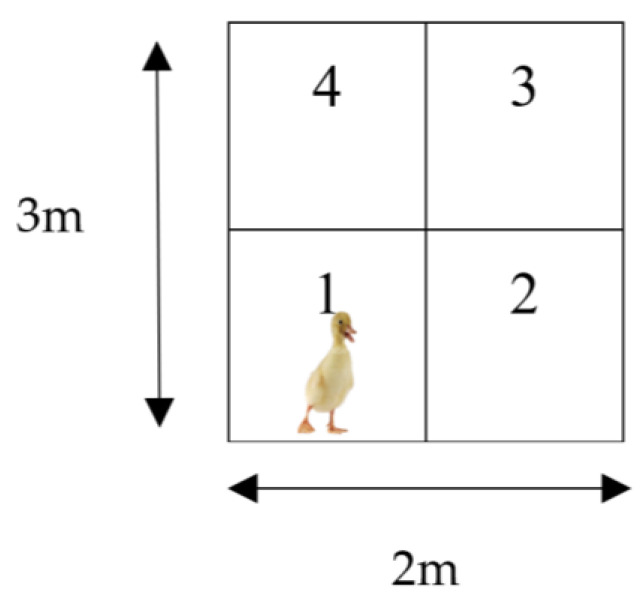
Experimental set-up for the 5 minutes open-field test.

**Table 1 animals-12-03479-t001:** Mean (±SEM) weight gains, feed intake, feed conversion ratios (FCR) by diet in ducks from days 1 to 21.

Item	Diet		
	Control	Probiotic	*p*-Value
Weight gain (g/b)	878 ± 22.9	824 ± 22	0.1054
Feed intake (g/b)	1119 ± 57.9	1064 ± 60.2	0.5249
FCR (feed/gain)	1.27 ± 0.04	1.29 ± 0.05	0.7710

Means are not significantly different (*p* > 0.05).

**Table 2 animals-12-03479-t002:** Mean (±SEM) serum corticosterone (CORT) (ng/mL) and heat shock protein (HSP) 70 (ng/mL) levels, and heterophil to lymphocyte ratios (HLR) by diet and crating treatment in ducks on day 21.

Treatment	CORT (ng/mL)	HSP70 (ng/mL)	HLR
**Diet** Control	24.3 ± 2.51 ^a^	3.30 ± 0.05 ^a^	0.87 ± 0.09 ^a^
Probiotic	18.6 ± 2.03 ^b^	3.17 ± 0.03 ^b^	0.72 ± 0.09 ^b^
**Crating treatment**			
Non-crated	13.6 ± 1.25 ^a^	3.17 ± 0.04 ^b^	0.43 ± 0.03 ^b^
Crated	29.3 ± 2.08 ^a^	3.29 ± 0.04 ^a^	1.24 ± 0.06 ^a^
**Source of variation**			
Diet (D)	0.0809	0.0182	0.0474
Crating treatment (CT)	<0.0001	0.0268	<0.0001
D × CT	0.0152	0.1779	0.6264

Means within a subgroup with no common superscripts differ at *p* < 0.05.

**Table 3 animals-12-03479-t003:** Mean (±SEM) serum corticosterone levels (ng/mL) where interaction between diet and crating treatment were significant in ducks on day 21.

Crating Treatment	Diet	
	Control	Probiotic
Non-crated	14.4 ± 1.61 ^a,y^	12.7 ± 1.98 ^a,y^
Crated	33.4 ± 1.99 ^a,x^	23.6 ± 3.49 ^b,x^

^a,b^ Means within a row with no common superscripts differ at *p* < 0.05. ^x,y^ Means within a column with no common superscripts differ at *p* < 0.05.

**Table 4 animals-12-03479-t004:** Mean (±SEM) serum levels of creatinine kinase (CK) (mg/mL), total protein (TP) (g/mL), triglyceride (TG) (mg/mL), glucose (GLU) (mg/mL), cholesterol (CHOL) (mg/mL), and lactate (LAC) (mg/mL) by diet and crating treatment in ducks on day 21.

Treatment	CK (mg/mL)	TP (g/mL)	TG (mg/mL)	GLU (mg/mL)	CHOL (mg/mL)	LAC (mg/mL)
**Diet**						
Control	1181 ± 85.7 ^a^	26.1 ± 0.47 ^a^	1.14 ± 0.12 ^a^	9.86 ± 0.13 ^a^	4.76 ± 0.10 ^a^	7.95 ± 0.29 ^a^
Probiotic	862 ± 40.6 ^b^	26.4 ± 0.53 ^a^	0.93 ± 0.08 ^b^	9.63 ± 0.16 ^a^	4.69 ± 0.14 ^a^	7.90 ± 0.26 ^a^
**Crating** **treatment**						
Non-crated	992 ± 58.67 ^a^	26.5 ± 0.46 ^a^	1.46 ± 0.09 ^a^	10.1 ± 0.10 ^a^	4.55 ± 0.09 ^b^	8.08 ± 0.25 ^a^
Crated	1065 ± 90.9 ^a^	26.1 ± 0.54 ^a^	0.61 ± 0.04 ^b^	9.41 ± 0.14 ^b^	4.89 ± 0.14 ^a^	7.80 ± 0.28 ^a^
**Source of variation**						
Diet (D)	0.0018	0.6265	0.0255	0.2722	0.6372	0.9866
Crating treatment (CT)	0.2938	0.6066	<0.0001	0.0002	0.0437	0.4899
D × CT	0.5191	0.0713	0.0714	0.6113	0.2084	0.0814

Means within a subgroup with no common superscripts differ at *p* < 0.05.

**Table 5 animals-12-03479-t005:** Mean (±SEM) tonic immobility duration (s) by diet and crating treatment in ducks on day 21.

Treatment	TI Duration
**Diet**	
Control	155 ± 24.8 ^a^
Probiotic	91.4 ± 19.4 ^b^
**Crating treatment**	
Non-crated	147 ± 21.2 ^a^
Crated	116 ± 24.8 ^a^
**Source of variation**	
Diet (D)	0.0482
Crating treatment (CT)	0.2127
D × CT	0.9976

Means within a subgroup with no common superscripts differ at *p* < 0.05.

**Table 6 animals-12-03479-t006:** Mean (±SEM) duration of latency to the first immobilisation (LA) (s), duration of mobility (DM) (s), number of zones visited (ZV), number of pecks directed to the floor or the walls (P), frequency of body shaking (BS), and number of vocalisation (V).

Treatment	LA (s)	DM (s)	ZV (no.)	P (no.)	BS (no.)	V (no.)
**Diet**						
Control	7.36 ± 2.57 ^a^	106 ± 14.72 ^a^	3.67 ± 0.11 ^a^	3.18 ± 0.62 ^a^	2.70 ± 0.39 ^a^	459 ± 32.9 ^a^
Probiotic	4.52 ± 1.14 ^a^	107 ± 16.749 ^a^	3.20 ± 0.20 ^b^	2.14 ± 0.53 ^a^	1.67 ± 0.34 ^b^	415 ± 32.2 ^a^
**Crating treatment**						
Non-crated	1.50 ± 0.53 ^b^	42.3 ± 6.24 ^b^	3.13 ± 0.20 ^a^	2.67 ± 0.67 ^a^	2.43 ± 0.27 ^a^	359 ± 24.8 ^b^
Crated	10.7 ± 2.37 ^a^	171 ± 13.2 ^a^	3.73 ± 0.11 ^b^	2.66 ± 0.50 ^a^	1.93 ± 0.45 ^a^	515 ± 33.4 ^a^
**Source of variation**						
Diet (D)	0.1558	0.9502	0.0356	0.2081	0.0298	0.3044
Crating treatment (CT)	<0.0001	<0.0001	0.0076	0.9750	0.2852	0.0005
D × CT	0.0005	0.9431	0.3601	0.0944	0.0003	0.6517

Means within a subgroup with no common superscripts differ at *p* < 0.05.

**Table 7 animals-12-03479-t007:** Mean (±SEM) duration of latency to the first immobilisation (LA) (s) and frequency of body shaking (BS) where interaction between diet and crating treatment were significant in ducks on day 21.

	LA		BS	
	Control	Probiotic	Control	Probiotic
**Crating treatment**				
Non-crated	0.57 ± 0.23 ^a,y^	2.58 ± 1.07 ^a,y^	2.07 ± 0.25 ^a,x^	2.80 ± 0.47 ^a,x^
Crated	19.25 ± 4.77 ^a,x^	6.07 ± 1.81 ^b,x^	3.33 ± 0.71 ^a,x^	0.53 ± 0.27 ^b,y^

^a,b^ Means within a row with no common superscripts differ at *p* < 0.05. ^x,y^ Means within a column with no common superscripts differ at *p* < 0.05.

## Data Availability

Data sharing not applicable. No new data were created or analyzed in this study. Data sharing is not applicable to this article.

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
