# Peer review of "Alleviation of Catching and Crating Stress by Dietary Supplementation of Bacillus subtilis in Pekin Ducks"

_animals, 2022, doi:10.3390/ani12243479_

Round 1
Reviewer 1 Report
The manuscript entitled “Alleviation of catching and crating stress by dietary supplementation of a probiotics (CLOSTAT) in Perkin ducks” provides some useful information in the field of animal sciences. Some comments of this manuscript are listed below.
Comments
1. Italicize all the scientific names present in text and in references section
2. In this journal, three to ten pertinent keywords need to be added, so that please add more keywords. Generally, keywords should contain words and phrases that suggest what the topic is about. Also include words and phrases that are closely related to your topic. Also use variant terms or phrases that readers are likely to use. The full forms of shortened words and abbreviations should be included as well.
3. Bacterial strain used as probiotic (Bacillus subtilis) and the dose for supplementing in the diet for ducks or poultry should be mentioned in Introduction and Discussion sections.
4. In Materials and methods, correct the CFU of probiotic, and clarify how to freshly prepare the diet containing B. subtilis. In addition, to avoid the loss of prebiotic due to storage time, the frequency of diet preparation should be mentioned.
5. Figure 1, the caption for a figure appears below the graphic.
6. Please present numerical data throughout the manuscript to 3 significant figures, e.g., 1.23, 12.3, 123, 1230.
7. t-test was applied for the data in Table 1 but this statistical tool was not described in section 2.7
8. L184 and onwards, rephrase the statement in text ‘diet ×crating treatment’ by ‘interaction between diet and crating treatment’ or other similar words
9. Lack of consistency in using ‘min’ or ‘minute’ and 'p-value' or 'P-value'. Please correct throughout the manuscript
10. Conclusion part should be able to stand alone, so that full words of each abbreviated term should be added.
11. The reference lists were not well checked and contain many errors, in both content and format. The reference errors found such as capital/lowercase letters and scientific names in the article titles.
Author Response
Dear Sir,
Please see the attachment.
Thank you

Reviewer 2 Report
mentation of a probiotic in Pekin ducks" demonstrated that four hours catching and crating of the ducks can cause stress and fear in the bird, and bacilli probiotics at 1 g/kg diet was beneficial in dampening the stress and fear responses; and also reduced corticosterone and HSP70. Use of the B. subtilis was also promising on tonic immobility duration and behaviors on bird. This result shows that use of probiotic in duck feed can assist the practical management of duck during catching and crating.
The paper contains novel data. Authors are recommend to not use the commercial name of probiotic in the title. Just ues probiotic, Bacillus subtilis.
Please double cross check your references numbers in the text and in the list.
Author Response

(The authors gave the same response as above.)
